# Intelligent Reflecting Surface-Assisted Physical Layer Key Generation with Deep Learning in MIMO Systems

**DOI:** 10.3390/s23010055

**Published:** 2022-12-21

**Authors:** Shengjie Liu, Guo Wei, Haoyu He, Hao Wang, Yanru Chen, Dasha Hu, Yuming Jiang, Liangyin Chen

**Affiliations:** 1School of Computer Science, Sichuan University, Chengdu 610065, China; 2Institute for Industrial Internet Research, Sichuan University, Chengdu 610065, China

**Keywords:** intelligent reflecting surface, physical layer, deep learning, secret key generation

## Abstract

Physical layer secret key generation (PLKG) is a promising technology for establishing effective secret keys. Current works for PLKG mostly study key generation schemes in ideal communication environments with little or even no signal interference. In terms of this issue, exploiting the reconfigurable intelligent reflecting surface (IRS) to assist PLKG has caused an increasing interest. Most IRS-assisted PLKG schemes focus on the single-input-single-output (SISO), which is limited in future communications with multi-input-multi-output (MIMO). However, MIMO could bring a serious overhead of channel reciprocity extraction. To fill the gap, this paper proposes a novel low-overhead IRS-assisted PLKG scheme with deep learning in the MIMO communications environments. We first combine the direct channel and the reflecting channel established by the IRS to construct the channel response function, and we propose a theoretically optimal interaction matrix to approach the optimal achievable rate. Then we design a channel reciprocity-learning neural network with an IRS introduced (IRS-CRNet), which is exploited to extract the channel reciprocity in time division duplexing (TDD) systems. Moreover, a PLKG scheme based on the IRS-CRNet is proposed. Final simulation results verify the performance of the PLKG scheme based on the IRS-CRNet in terms of key generation rate, key error rate and randomness.

## 1. Introduction

With the development of wireless communication technologies, more mobile devices will be connected to wireless systems [1]. Secure problems must be taken seriously because of the broadcast and openness of the wireless channel [2]. Traditionally, encryption algorithms, including symmetric key cryptography and asymmetric key cryptography, have been used to ensure the communication security [3]. However, traditional security schemes rely on public key infrastructures and complex encryption algorithms to manage secret keys [4], and they are not suitable for the Internet of Things (IoT) networks because IoT devices have constrained computational ability and resources. To address this issue, the physical layer key generation (PLKG), which exploits the inherent randomness of wireless fading channels, has become a promising technology to generate a shared secret key between wireless devices. The feasibility of PLKG relies on three principles, i.e., temporal variation, channel reciprocity, and spatial decorrelation [5,6].

Some works [7,8,9,10] have studied the process of PLKG, but they lack consideration of the signal interference existing between legitimate communication parties. In practice, the signal is susceptible to blockages such as a building or a wall. For these non-ideal communication environments, additional random signals and relay nodes [11,12] were introduced to improve the performance of PLKG schemes, while these methods can cause high power consumption and computation complexity, so they are not applicable to resource-constrained IoT devices. Therefore, how to design an efficient PLKG scheme in scenarios where signal interference exists is still an open question.

Nowadays, the reconfigurable intelligent reflecting surface (IRS) has been regarded as an intrinsic component in future wireless systems [13]. Some works [14,15,16,17,18,19,20] have employed the IRS to assist PLKG, and some of them considered SISO systems, while others considered MIMO systems. If we study PLKG in MIMO systems, how to reduce the overhead of channel reciprocity extraction is a crucial problem, especially for IoT devices with constrained resources. In addition, much prior knowledge is required in these works. Deep learning can alleviate these problems, as it is a meaningful technology that can bring promising applications to the physical layer [21], i.e., deep learning-based block-structured communications [22], signal recognition [23], channel estimation [24] and CSI feedback denoising [25]. For non-ideal communication environments, current PLKG schemes cannot achieve both low computation complexity and excellent performance.

In this paper, we introduce an IRS into the communication environment where signal interference exists. IRS has emerged as a promising technology to improve communication qualities through adjustments [15]. IRSs comprise massive numbers of nearly-passive elements interacting with incident signals [13], and the location of these elements can be adaptively placed. We can manipulate the wireless channel by adjusting the reflection coefficients of IRS continuously or discretely with low power consumption, i.e., phase, amplitude, frequency and even polarization [26,27]. Moreover, by exploiting deep learning networks, we design IRS-CRNet to extract reciprocal channel features in the orthogonal frequency division multiplexing (OFDM) TDD systems with low overhead. Then an efficient PLKG scheme based on IRS-CRNet is proposed for MIMO systems. The main contributions of this paper are as follows.
We introduce an IRS into scenarios where blockages exist between legitimate communication parties to assist PLKG. Then, we construct the hybrid channel function and achieve an optimal achievable rate by adjusting phase shifts.We design the IRS-CRNet that can efficiently learn the reciprocity from the channel state information (CSI). Without any prior knowledge and much computational overhead, IRS-CRNet trained with a hybrid loss function can extract the channel features with high reciprocity, which can be used for generating the initial key directly.Based on the IRS-CRNet, we propose a novel IRS-assisted PLKG scheme for TDD systems. Experimental simulation results show that the performance of this scheme is excellent in terms of three metrics, including key generation rate, key error rate and randomness.

The rest of this paper is organized as follows. The second part presents the related works. In the third part, we construct the channel function and the reciprocal channel features-learning model. A novel IRS-assisted PLKG scheme based on deep learning is also proposed in this part. In the fourth part, the simulation results are presented to evaluate the performance of this scheme. Finally, the last part concludes this paper with a summary of our work.

## 2. Related Work

### 2.1. PLKG without IRS

Different metrics in the wireless communication system have been used to assist PLKG. Zhan et al. [10] proposed a PLKG scheme using the multi-level discrete wavelet transform to enhance the availability of key generation in real environments. Mathur et al. [28] extracted the RSS information of the wireless channel and exploited it to assist PLKG. The RSS-based methods cannot achieve high KGR and sufficient randomness, and more works [7,8,9] exploiting CSI have achieved better performance.

Many works have studied PLKG in MIMO systems so far. Li et al. [20] proposed a multiuser secret key generation in massive MIMO wireless networks and focused on the sum secret ket rate maximization. Jiao et al. [19] exploited new channel characteristics, including AoA (angle to arrival) and AoD (angle to departure), to generate the secret key. Jorswieck et al. [17] studied the secret key rate for a model with a MIMO channel and showed the impact of the statistic of the MIMO channel. Furqan et al. [18] developed a key generation method based on channel quantization with singular value decomposition. Moreover, some works consider protecting PLKG from attacks. Mitev et al. [29] analyzed the rejection and reactive jamming attacks in MIMO PLKG systems, and they showed a pilot randomization scheme can reduce injection attacks to jamming attacks and used a game-theoretic approach to deal with jamming attacks.

Deep learning has been applied to the process of the physical layer key generation. Zhang et al. [3] applied deep learning for PLKG in FDD for the first time, they first proved the existence of the band feature mapping function and proposed a key generation neural network (KGNet), the results turned out to be good. He et al. [30] designed the Channel Reciprocity Learning Net (CRLNet) to learn the channel reciprocity features of the wireless channels, and the CRLNet-based key generation scheme showed good performance. Zhou et al. [31] proposed a PLKG scheme combining the autoencoder and the multi-task learning, and this scheme can extract the reciprocal features from the weakly correlated channel.

All works mentioned above only focused on the relatively ideal wireless communication environments in which no blockage exists between transmitters and receivers. However, blockages exist in most wireless communication systems in practice, and they can interfere with the signal severely. To solve this problem, IRS can be exploited to assist PLKG.

### 2.2. PLKG with IRS

A few works have studied applying IRS to the PLKG scheme, and combining IRS with the conventional transmission control can potentially bring about performance gain compared with wireless networks without IRS [32]

Ji et al. [16] formulated the minimum achievable secret key capacity for an IRS acting as a passive beamformer in the presence of multiple eavesdroppers, and they designed an SDR-SCA optimization algorithm to maximize the minimum achievable secret key capacity for the worst-case eavesdropper. Lu et al. [33] proposed a key generation protocol with the aid of IRS to boost the secret key rate in quasi-static environments. Ji et al. [34] studied IRS-assisted PLKG schemes in scenarios with a single user and multiple eavesdroppers. Li et al. [35] introduced a multiuser secret key generation scheme that capitalizes on the presence of IRS, and it achieved a high sum secret key rate. Liu et al. [14] proposed a deep reinforcement learning approach to boost the secret key generation in an IRS-assisted system. Taha et al. [13] developed a deep learning-based solution in which IRS learns how to interact with the incident signal to increase the achievable rate.

However, current works mainly focus on SISO systems, and deep learning has not been effectively used for this application. In this paper, we propose an IRS-assisted PLKG scheme with deep learning to generate the shared consistent keys for TDD MIMO systems.

## 3. Materials and Methods

### 3.1. Channel Model

The process of PLKG involves two legitimate communication parties, called Alice and Bob, as well as an eavesdropper, called Eve. As shown in Figure 1, we assume a wireless communication scenario with a blockage. In this paper, we regard the base station as Alice and the user device in the user grid area as Bob, so either User_A or User_B can be regarded as Bob. Meanwhile, compared with User_A, the signal strength User_B receives should be smaller because the signal User_B receives is blocked more severely, and we apply an IRS to achieve the signal strength gain. By introducing an IRS into the scenario with a blockage, we combine the direct channel and the reflecting channel in a TDD system to construct the channel function.

In the OFDM-based TDD system with *K* subcarriers we adopt, we assume that IRS and the base station have both *M* antennas, and the user device has 8 antennas. Note that the base station can be either the transmitter or the receiver, as can the user device. For the reflecting channel, we define $H_{B,R}$ as the M×M×K downlink channel from the base station to IRS, $H_{R,U}$ as the 8×M×K downlink channel from IRS to the user, $H_{U,R}$ as the M×8×K uplink channel from the user to IRS, and $H_{R,B}$ as the M×M×K uplink channel from IRS to the base station. For the direct channel, we define $H_{B,U}$ as the 8×M×K downlink channel from the base station to the user, and $H_{U,B}$ as the M×8×K uplink channel from the user to the base station. Note that all these data are complex matrices. $H_{B,R,k}$ represents the channel information of $H_{B,R}$ at the *k*th subcarrier; similarly, we can obtain other channel information. Taking the user device as the receiver, then we can obtain the received signal at the *k*th subcarrier as
(1)$$H_{U,k}=\left(H_{R,U,k}\Phi H_{B,R,k}\right)⊙T_{B,k}+H_{B,U,k}⊙T_{B,k}+N_{k}$$
taking the base station as the receiver, we obtain a similar equation as
(2)$$H_{B,k}=\left(H_{R,B,k}\Phi H_{U,R,k}\right)⊙T_{U,k}+H_{U,B,k}⊙T_{U,k}+N_{k}$$
where Φ denotes a M×M IRS interaction matrix, which represents how IRS interacts with the incident signal. $T_{B,k}$ and $T_{U,k}$ denote the transmitted signal over the *k*th subcarrier and satisfy $E[|T_{\alpha ,k}{|}^{2}]=\frac{P_{T}}{K}$ where α∈{B,U}. $P_{T}$ represents the total transmitting power. $N_{k}$ is the received noise and satisfies $N_{k}∼N_{C}\left(0,\sigma_{N}^{2}\right)$.

The achievable signal rate can be expressed as
(3)$$R=\frac{1}{K}\sum_{k=1}^{K}\log_{2}\left(1+SNR|\left(H_{R,U,k}\Phi H_{B,R,k}\right){|}^{2}\right)$$
where $SNR=\frac{P_{T}}{K\sigma_{N}^{2}}$ denotes the signal-to-noise ratio. To maximize the achievable rate at the receiver, we need to design an optimal interaction matrix Φ, which can be expressed as
(4)$$\begin{array}{cc} \Phi & =(\phi_{0},\;\phi_{1},\;\phi_{2},\;\ldots \;,\;\phi_{M-1})\\ \; & =\left(\begin{array}{ccccc} \psi_{0} & 0 & 0 & \ldots & 0\\ 0 & \psi_{1} & 0 & \ldots & 0\\ 0 & 0 & \psi_{2} & \ldots & 0\\ \ldots & \ldots & \ldots & \ldots & \ldots \\ 0 & 0 & 0 & \ldots & \psi_{M-1} \end{array}\right) \end{array}$$
where Φ has a diagonal structure and $\phi_{i}$ denotes a beamforming vector. We assume locations of different devices are shown in Figure 2. $P_{B}$, $P_{R}$ and $P_{U}$ represent the base station, the IRS and the user device, respectively. $P_{R}^{\prime }P_{R}^{\prime \prime }$ represents an element of IRS, and we lengthen it to make it clear. $P_{R}O^{\prime }$ is the angular bisector of $∠P_{B}P_{R}P_{U}$, and right angles include $∠P_{B}OP_{R}$, $∠O^{\prime }P_{R}P_{R}^{\prime \prime }$, $∠OP_{R}O^{\prime \prime \prime }$ and $∠P_{U}O^{\prime \prime }P_{R}$. According to the geometrical model, we can obtain a theoretically optimal reflecting angle β. We can deduce angles as
(5)$$\alpha =\arctan\frac{\left|x_{R}-x_{B}\right|}{\left|y_{R}-y_{B}\right|}$$
(6)$$\gamma =\arctan\frac{\left|x_{U}-x_{R}\right|}{\left|y_{U}-y_{R}\right|}$$
(7)$$\beta =max\left(\alpha ,\gamma \right)-\frac{\alpha +\gamma }{2}$$
to maximize the achievable rate, and we need to adjust the phase of every element of IRS according to Equation (Equation 7). Assuming the location of the *i*th element of IRS as $\left(x_{R,i},y_{R.i}\right)$, an extensive equation can be expressed as
(8)$$\psi_{i}=max\left(\arctan\frac{\left|x_{R,i}-x_{B}\right|}{\left|y_{R,i}-y_{B}\right|},\arctan\frac{\left|x_{U}-x_{R,i}\right|}{\left|y_{U}-y_{R,i}\right|}\right)-\frac{\arctan\frac{\left|x_{R,i}-x_{B}\right|}{\left|y_{R,i}-y_{B}\right|}+\arctan\frac{\left|x_{U}-x_{R,i}\right|}{\left|y_{U}-y_{R,i}\right|}}{2}$$
we can obtain a theoretically optimal interaction matrix Φ according to Equation (8).

Moreover, we assume the location of the eavesdropper, Eve, is more than half of the wavelength away from Alice and Bob. Then the channel of Eve is irrelevant to the channel of legitimate users, so Eve cannot deduce the channel of legitimate users and threaten the key generation.

### 3.2. Channel Reciprocal Features Extraction Model

In this paper, we focus on exploiting IRS to assist PLKG with deep learning in a non-ideal scenario where signal interference exists between legitimate communication parties, as shown in Figure 1. Channel reciprocity could be affected by many factors. Though the frequency of the downlink ($f_{d}$) and the frequency of the uplink ($f_{u}$) are normally assumed to be exactly the same in an OFDM-based TDD system, a small frequency offset exists between $f_{d}$ and $f_{u}$, and it may cause non-ideal reciprocity. Additionally, the noise that existed in the channel response may decrease the reciprocity further. To exftract the reciprocal features from the wireless channel with deep learning, we propose a model called IRS-CRNet to learn the reciprocity of the channel response.

The architecture of IRS-CRNet is shown in Figure 3. According to Equation (Equation 1), we propose an optimized Encoder and Decoder. Encoder is designed to extract the reciprocal components $R_{\alpha }$ from the original channel response $H_{\alpha }$; meanwhile, Decoder is exploited to keep the same dimensionality of $H_{\alpha }$ with maintained high reciprocity. The detailed hyper-parameters of each non-linear layer that we implement in Encoder are shown in Table 1. After the non-linear process, we flatten the tensor and send it to the fully-connected linear layers, and the numbers of fully-connected layers are 64, 128 and 64, respectively. The detailed hyper-parameters of Decoder are shown in Table 2.

In the training phase, we propose a hybrid loss function, which is expressed as
(9)$$Loss_{h}=L_{1}+L_{2}$$
where the loss function consists of two parts. $L_{1}$ is the mean squared loss (MSE) of $R_{U}$ and $R_{B}$, representing the quadratic sum of the deviation between the downlink channel response and the uplink channel response. $L_{1}$ is is expressed as
(10)$$L_{1}=\frac{1}{R\times T}\sum_{i=0}^{R\times T-1}{\left(R_{U,i}-R_{B,i}\right)}^{2}$$
where *R* and *T* denote the numbers of antennas of the receiver and the transmitter, respectively. We adopt MSE to make our model learn the reciprocity; however, MSE is not suitable for dealing with the abnormal data because it could give more weight to them. To solve this problem, we introduce another loss function $L_{2}$ called Log-Cosh, which is expressed as
(11)$$L_{2}=\frac{1}{R\times T}\sum_{i=0}^{R\times T-1}log\left(cosh\left(R_{U,i}-R_{B,i}\right)\right)$$

Compared with MSE, Log-Cosh is less susceptible to abnormal data and makes up for the deficiency of MSE.

### 3.3. PLKG Process

In this paper, we propose a novel IRS-assisted PLKG scheme with deep learning. The scheme contains the following steps.
Channel probing: Conventionally, two legitimate communication parties, i.e., Alice and Bob, send a pilot signal synchronously, and then we can estimate the channel state information (CSI). However, after introducing an IRS, we need to consider the reflecting channel. According to Equation (Equation 1), the combined channel response can be expressed as
(12)$$H_{\alpha }=H_{r}+H_{d}+N_{t}$$
where α∈{B,U}. $H_{r}$ and $H_{d}$, respectively, represent the reflecting channel response and the direct channel response of the wireless channel. $N_{t}$ represents the total noise of the wireless channel.Reciprocal Channel Features Extraction: Though we adopted the TDD system, the actual reciprocity is not great enough to be used directly to generate the key. We need to extract the reciprocal channel features from the estimated CSI after combining the reflecting and direct channel according to Equation (Equation 12). Because of the significant amount of prior knowledge required, it is difficult to extract the reciprocity by theoretical equations. Therefore, we exploit the IRS-CRNet to extract the reciprocal channel features.Quantization: We intend to convert the channel features to a binary bit sequence with high key generation rate, low key error rate and sufficient randomness. First, we need to preprocess the original channel matrix over each subcarrier with the method below
(13)$${\bar{H}}_{\alpha ,k}=Flatten\left(\left[Real\left(H_{\alpha ,k}\right)\;\;Imag\left(H_{\alpha ,k}\right)\right]\right)$$
where Real(∗) represents the real part of ∗, Imag(∗) represents the image part of ∗, and Flatten(∗) converts * from a matrix to a vector. Then, we normalize the vector to keep the value of each element of the vector in the range of 0 to 1, and we convert each element according to the equation below
(14)$${\bar{H}}_{\alpha ,k,i}^{\prime }=\frac{{\bar{H}}_{\alpha ,k,i}-min\left({\bar{H}}_{\alpha ,k}\right)}{max\left({\bar{H}}_{\alpha ,k}\right)-min\left({\bar{H}}_{\alpha ,k}\right)}$$
where ${\bar{H}}_{\alpha ,k,i}$ represents the *i*th element of ${\bar{H}}_{\alpha ,k}$, min(∗) represents the minimum value of ∗ and max(∗) represents the maximum value of ∗. Finally, we propose a multi-level quantization method based on the percent point function (PPF), which is the inverse of the cumulative distribution function, to quantize the normalized sequence. The process of quantization is expressed as Algorithm 1.Information Reconciliation and Privacy Amplifying: The mismatched bits in the initial key can be corrected by information reconciliation to reduce KER. Information reconciliation can be realized by many protocols, i.e., BCH code [36], ECC [37], Cascade [38], and Golay code [39]. The privacy amplifying phase mostly exploits the hash function to convert the corrected key sequence with the information reconciliation to a shorter secret key, which can be used directly.

**Algorithm 1** The multi-level quantization algorithm based on PPF**Input:** The normalized feature vector H¯α,k′; the quantization factor δ with a default value: 10−2**Output:** The quantized bit sequence Qα;
  1:Calculate the mean μ and the standard deviation σ of the feature vector H¯α,k′;  2:Use PPF of the scipy.stats library (ppf) to confirm the benchmarks;  3:low1=ppf(0.25−δ,loc=μ,scale=σ);  4:high1=ppf(0.25+δ,loc=μ,scale=σ);  5:low2=ppf(0.75−δ,loc=μ,scale=σ);  6:high2=ppf(0.75+δ,loc=μ,scale=σ);  7:Initialize Qα as an empty list;  8:**for**iinrange(len(H¯α,k))**do**  9:  **if** H¯α,k,i′>=low1
**and**
${\bar{H}}_{\alpha ,k,i}^{\prime }<=high_{1}$ **then**10:   $Q_{\alpha }.append\left(0\right)$11:  **else**12:   **if** ${\bar{H}}_{\alpha ,k,i}^{\prime }>=low_{2}$
**and**
${\bar{H}}_{\alpha ,k,i}^{\prime }<=high_{2}$ **then**13:    $Q_{\alpha }.append\left(1\right)$14:   **else**15:    $Q_{\alpha }.append\left(-1\right)$16:   **end if**17:  **end if**18:**end for**19:return $Q_{\alpha }$.


## 4. Experimental Results

### 4.1. Simulation Setup

We adopted the DeepMIMO [40] dataset to generate the channels and the dataset we needed based on the outdoor ray-tracing scenario ‘O1’ (Outdoor 1). The crucial part that we need to focus on is shown in Figure 4. The DeepMIMO dataset parameters we adopted are summarized in Table 3.

According to the OFDM TDD system we adopted, we set a constant operating frequency at 3.4 GHz. Active users from R551 to R1100 are exploited to generate the training dataset with a total of 99,550 channel responses, while active users from R1101 to R1200 are exploited to generate the testing dataset with a total of 18,100 channel responses. The training dataset is divided into 550 files, and the testing dataset is divided into 100 files. Each file contains information of 181 CSI matrices. In addition, hyper-parameters used to train IRS-CRNet are given in Table 4, and the input size of IRS-CRNet is set to (64,1,8,8).

### 4.2. Performance Metrics of PLKG

We use the following metrics to evaluate the performance of the PLKG scheme.
Key generation rate (KGR): In this paper, this metric is slightly different from KGR mentioned in other works. We define KGR as
(15)$$KGR=\frac{N_{bits}}{N_{rec}\times T_{trans}\times N_{sub}}$$
where $N_{bits}$ denotes the number of bits the initial key contains. $N_{rec}$, $N_{trans}$ and $N_{sub}$ denote the number of receiver antennas, transmitter antennas and subcarriers, respectively. According to Equation (Equation 15), we normalize and keep the value of KGR in the range of 0 to 1.Key error rate (KER): It is defined as the number of error bits divided by the number of total bits [3].Randomness: The randomness of the initial key generated based on the PLKG scheme is important to maintain security. We use a test suit based on The National Institute of Standards and Technology (NIST) [42] to evaluate the randomness of the initial key.

### 4.3. Performance of Achievable Rate

To verify the effectiveness of the introduced IRS, we take the bottom user device at R1200 in Figure 4 for a test. The performance of the IRS and the interaction matrix Φ according to Equation (8) is evaluated by the achievable rate. In Figure 5, we compare the achievable rate under different conditions. We first obtain the ideal achievable rate when IRS has different numbers of antennas, including 1×4×4, 1×8×8 and 1×16×16. Then we obtain the actual achievable rate when IRS is introduced or not. Compared with the condition when IRS is not introduced, the achievable rate when IRS is introduced is apparently better and very close to the ideal one. When SNR exceeds 20 dB, we can obtain a steady and relatively ideal achievable rate.

### 4.4. Performance of IRS-CRNet

In this section, we evaluate the performance of different models. We compare IRS-CRNet with three other networks shown as follows.
AutoEncoder [21]: It is a simple autoencoder model which consists of an encoder and a decoder. It inspires a new way of thinking as an end-to-end reconstruction optimization task.DNN [43]: It is designed for channel calibration in generic massive MIMO systems. It has the potential in many parameter estimation problems for communications. It has the multilayer structure with three hidden fully connected layers.KGNet [3]: It is designed for frequency band feature mapping to construct reciprocal channel features between legitimate communication parties in SISO FDD systems. It has a multilayer structure with four hidden, fully connected layers.

The result of loss functions during the training phase is an important metric of the performance of models. Figure 6 shows how the results of $Loss_{h}$ of four models change as the training epoch grows. Every model is trained for the same time based on the same training dataset with a constant of 99,550 training sets. After 100 training epochs, all four models achieve stable results of $Loss_{h}$.

In Figure 7, we compare the performance of four models on the testing dataset through $Loss_{h}$ and MSE, and the effectiveness of $Loss_{h}$ is verified. Compared with models trained with MSE, the models trained with $Loss_{h}$ obtained better results. According to Figure 6 and Figure 7, our model performs better than other three models in both the training phase and testing phase. Moreover, though the number of the hidden layers in KGNet is bigger than the number in DNN, KGNet does not achieve appropriate performance gain. We can tell that the increased numbers of hidden layers cannot directly improve the performance of a model.

The original CSI matrices are complex matrices, and we evaluate the performance of IRS-CRNet on the real part of CSI matrices. In Figure 8a, we compare downlink channel features with uplink channel features; these features cannot be directly used for key generation because of many nonreciprocal features. We exploit the trained IRS-CRNet to extract the reciprocal features existing in both downlink and uplink channel responses. Channel features predicted by IRS-CRNet with great reciprocity are shown in Figure 8b.

### 4.5. Performance of PLKG Scheme Based on IRS-CRNet

We implement some experiments and use the performance metrics mentioned before to evaluate the performance of the PLKG scheme we designed. We generate the initial key based on the testing dataset with 18,100 sets. In Figure 9, we compare the performance of KGR and KER with different values of the quantization factor δ, i.e., 0.01, 0.05, 0.1 and 0.2. The results show that KGR becomes bigger and the KER becomes smaller with the value of δ growing. In addition, we add complex Gaussian noise in the range of 0–40 dB with a 5 dB step. Figure 9 also shows how KGR and KER change when SNR changes—KGR increases and KER decreases significantly when SNR increases. Moreover, to keep KER at an ideally low level, we can set different values of δ under different SNRs.

In this paper, we focus on the scenario where blockages exist between legitimate communication parties, and we introduce an IRS to assist the key generation. We need to prove the effectiveness of the introduced IRS. For the scenario without IRS, we do not consider the reflecting channel in Equation (Equation 1), but we keep other phases the same. We use the same training dataset to train our model, and we test the results on the same testing dataset. In Figure 10, we can see the performance is much better when the IRS is introduced. Moreover, the more antennas IRS has, the more bits the initial key has.

In Figure 11, we compare the performance of KGR when we set different numbers of antennas that IRS has. KGR is not acceptable when IRS is not introduced, and KGR only slightly increases when SNR increases because of limited channel features. However, the constructed channel responses contain more detailed information when IRS is introduced, and IRS-CRNet can learn great reciprocity to help improve KGR. The performance of secret keys generated by different models when IRS with 1×16×16 elements is introduced is shown in Table 5. Compared with prior works, IRS-CRNet apparently achieves better performance.

To verify the randomness of the generated initial keys, we used the Github repository randomness_testsuite as the randomness testing suite, which is implemented based on a NIST statistical test suite. The quantization factor was set to 0.01, and we generated a secret key based on the PLKG scheme. Then we took the bit sequence as the input of randomness_testsuite. The test results are shown in Table 6, and all the test types passed the randomness test.

### 4.6. Overhead Analysis

The models were implemented on a computer with AMD Ryzen 7 4800U, 16 GB RAM and Windows 11 Professional 64-bit operating system. Pytorch 1.12 was employed as the deep learning framework. To analyze computational overhead, the number of antennas that the base station and IRS exploit is 1×8×8, and the training time and CPU load of different models are shown in Table 7. In addition, the time of extracting reciprocity from a pair of downlink/uplink channel responses was calculated. In general, the training of IRS-CRNet does not need the high-performance GPU, and the trained model can effectively extract reciprocity; thus, the overhead is quite acceptable.

## 5. Conclusions

In this paper, we consider an IRS-introduced wireless communication scenario in which blockages exist between legitimate communication parties and propose an efficient method to assist PLKG. First, we construct the channel function combining the direct channel and the reflecting channel. Then a theoretically optimal interaction matrix is proposed to approach the optimal achievable rate. Moreover, we design the IRS-CRNet that can learn reciprocity from channel state information (CSI) matrices in the OFDM TDD MIMO systems. Based on the IRS-CRNet, we propose an efficient PLKG scheme for TDD systems. Finally, we implement sufficient experiments. The simulation results demonstrate that the introduced IRS indeed contributes to the key generation, the IRS-CRNet achieves excellent performance in the reciprocity-learning phase, and the PLKG scheme achieves high KGR, low KER and sufficient randomness.

In future work, on the one hand, we will study how to optimize the performance of IRS-assisted key generation in non-ideal scenarios with legitimate mobile communication users. On the other hand, we will extend this scheme to PLKG for massive TDD-MIMO systems. Moreover, how to effectively protect key generation from active and passive attacks could be meaningful research.

## Figures and Tables

**Figure 1 sensors-23-00055-f001:**
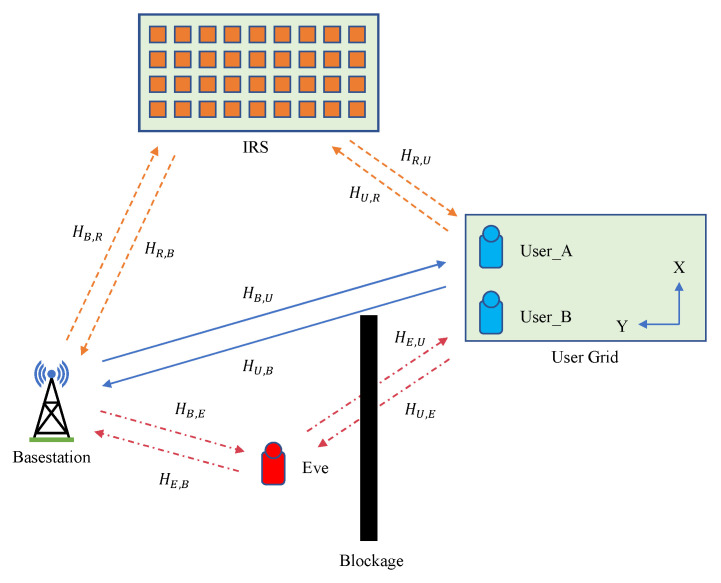
The wireless communication scenario with an IRS introduced. A blockage exists between the base station and user devices, including User_A and User_B, and interferes with the wireless signal. The received signal at User_B is interfered with more severely compared with User_A.

**Figure 2 sensors-23-00055-f002:**
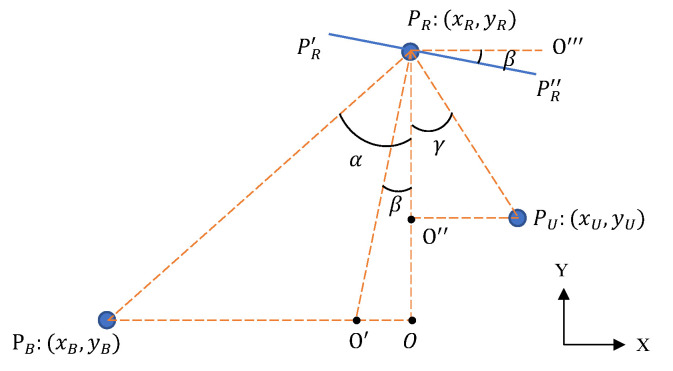
Assumed locations of the base station, the IRS and the user device.

**Figure 3 sensors-23-00055-f003:**
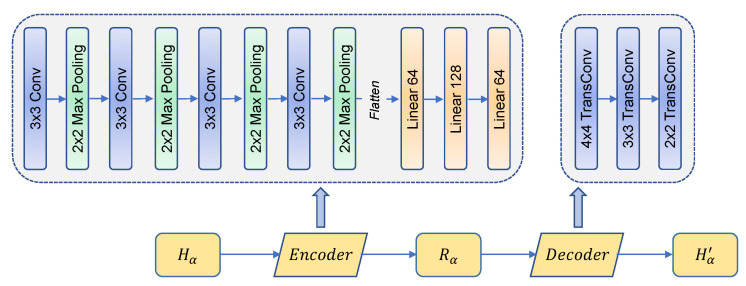
Structure of IRS-CRNet. Note that $H_{\alpha }\in \left\{H_{B},H_{U}\right\}$.

**Figure 4 sensors-23-00055-f004:**
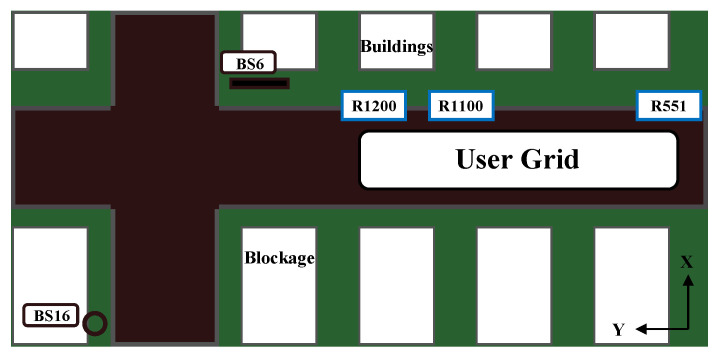
This figure represents a part of the O1 scenario. The sixth base station (BS6) is acting as an IRS to reflect the signal from a transmitter to a receiver. BS16 is a transmitter and also a receiver, as is every user device in the User Grid from row 551 (R551) to row 1200 (R1200). Every row contains 181 user devices. The building across the road from BS16 is acting as a blockage.

**Figure 5 sensors-23-00055-f005:**
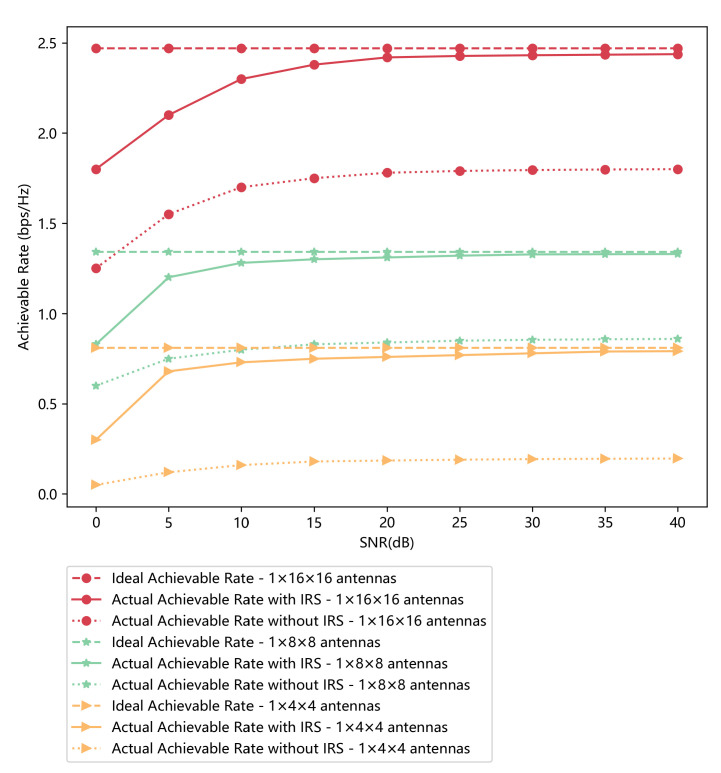
The achievable rate comparison under different conditions when the IRS has different numbers of elements.

**Figure 6 sensors-23-00055-f006:**
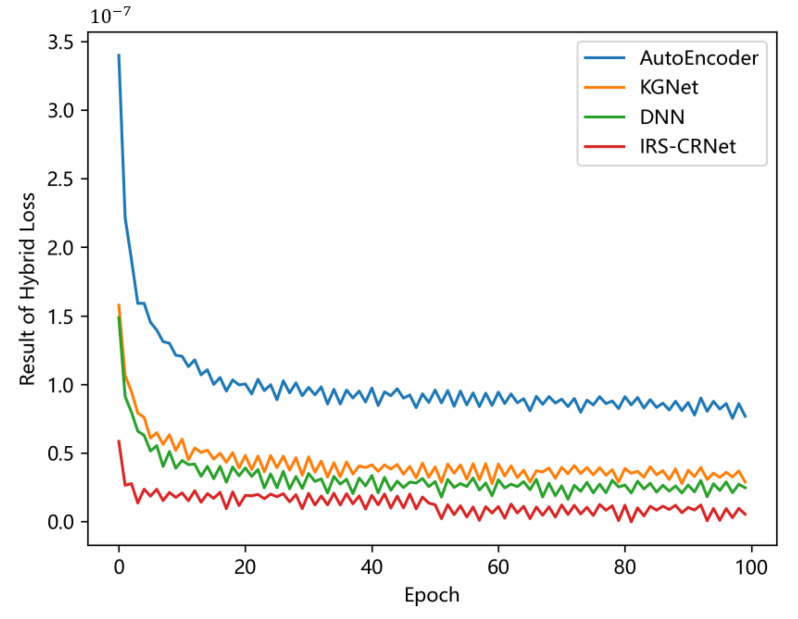
The result of $Loss_{h}$ of four different models versus the epoch of the training process.

**Figure 7 sensors-23-00055-f007:**
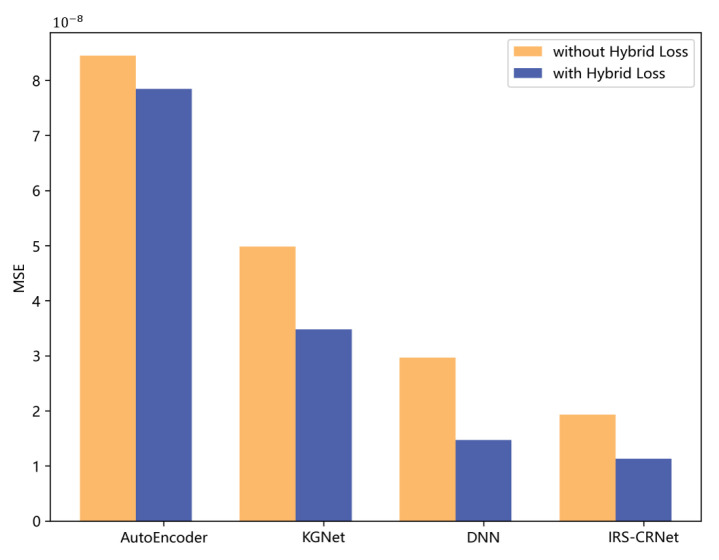
The MSE of four different models trained with $Loss_{h}$ and without $Loss_{h}$.

**Figure 8 sensors-23-00055-f008:**
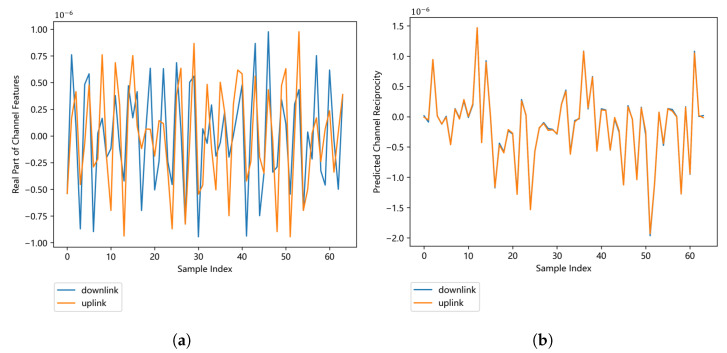
Comparison between the original channel features and the predicted channel features. (**a**) Channel response that contains reciprocal and nonreciprocal components. (**b**) Channel response that mostly contains predicted reciprocity extracted by IRS-CRNet.

**Figure 9 sensors-23-00055-f009:**
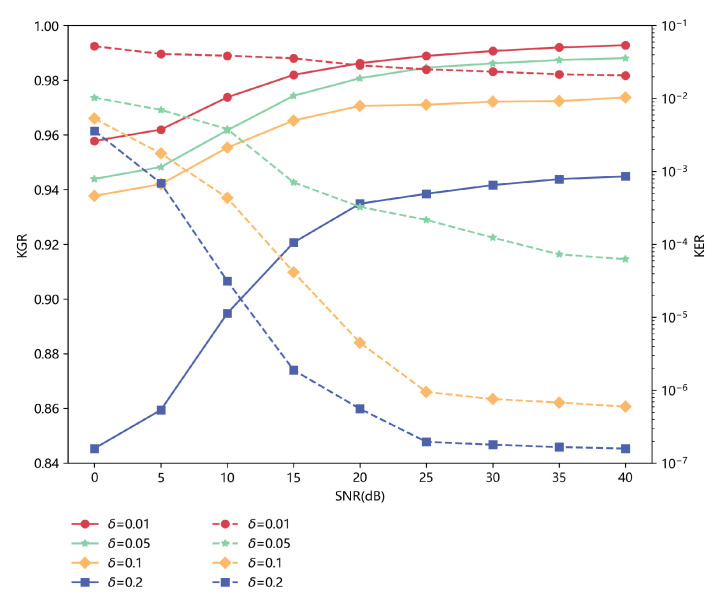
The KGR and KER of the initial key with different values of the quantization factor δ based on the IRS-CRNet versus SNR. The IRS-CRNet is trained with the training dataset with 99,550 sets of 0 dB. The solid line represents KGR, and the dashed line represents KER.

**Figure 10 sensors-23-00055-f010:**
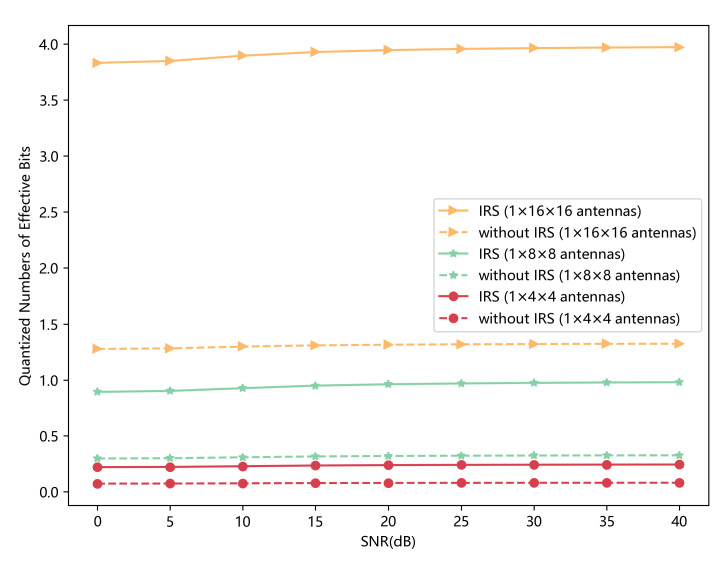
Quantized numbers of effective bits that initial keys have with different IRS parameters versus SNR. The scenario with an IRS having 1×8×8 antennas is treated as a benchmark. Note that δ is set to 0.01.

**Figure 11 sensors-23-00055-f011:**
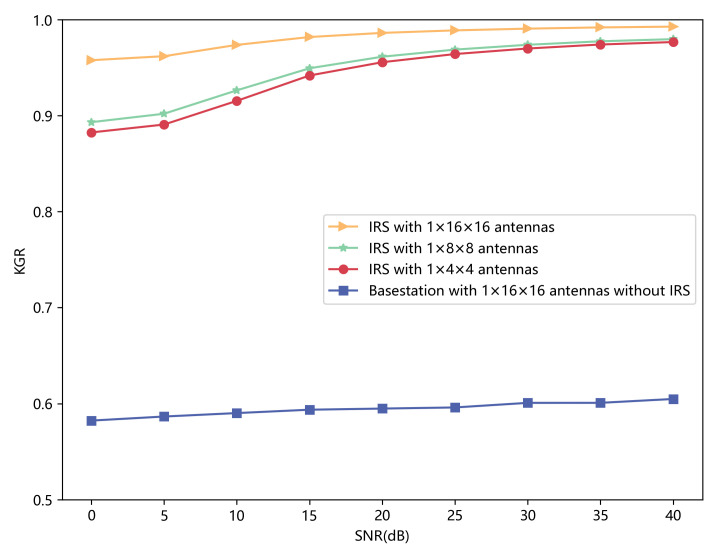
KGR of the initial key with a different number of antennas versus SNR.

**Table 1 sensors-23-00055-t001:** Details of different non-linear layers of Encoder.

Layer	Type	Kernel Size	Stride	Padding
1	Conv	(3,3)	1	0
2	Max Pooling	(2,2)	1	1
3	Conv	(3,3)	1	0
4	Max Pooling	(2,2)	1	1
5	Conv	(3,3)	1	0
6	Max Pooling	(2,2)	1	1
7	Conv	(3,3)	1	0
8	Max Pooling	(2,2)	1	1

**Table 2 sensors-23-00055-t002:** Details of layers of Decoder.

Layer	Type	Kernel Size	Stride	Padding
1	TransConv	(4,4)	1	1
2	TransConv	(3,3)	1	0
3	TransConv	(2,2)	1	0

**Table 3 sensors-23-00055-t003:** The adopted DeepMIMO dataset parameters.

DeepMIMO Dataset Parameters	Value
Active base stations (BSs)	6, 16
Active users (training)	from R551 to R1100
Active users (testing)	from R1101 to R1200
Number of BS antennas	(x,y,z)=(1,4,4);(1,8,8);(1,16,16)
Antennas spacing	0.5
Operating frequency	3.4 GHz
Number of OFDM subcarriers	512
OFDM limit	64
OFDM sampling factor	1
Number of paths	10

**Table 4 sensors-23-00055-t004:** Parameters for the IRS-CRNet.

Parameter	Value
Optimization	ADAM [41]
Exponential decay rates for ADAM: $\left(\tau_{1},\tau_{2}\right)$	(0.9, 0.999)
Learning rate	10${}^{-2}$
Batch size	64
Number of epochs	100
Number of training samples	99,550
Number of testing samples	18,100

**Table 5 sensors-23-00055-t005:** Performance of different models.

Different Models	KGR	KER
AutoEncoder [21]	0.9437	5.3457 × 10${}^{-2}$
KGNet [3]	0.9735	6.7642 × 10${}^{-3}$
DNN [43]	0.9880	9.5565 × 10${}^{-6}$
IRS-CRNet	0.9936	1.8265 × 10${}^{-7}$

**Table 6 sensors-23-00055-t006:** NIST statistical test results.

Test Type	*p*-Value	Result
Approximate Entropy	0.8562	Random
Block Frequency	0.7172	Random
Cumulative Sums	0.9969	Random
Discrete Fourier Transform	0.9668	Random
Frequency	0.7172	Random
Ranking	0.8014	Random
Runs	0.3585	Random
Serial	0.4990	Random

**Table 7 sensors-23-00055-t007:** Computational overhead of different models.

Networks	Training Time	CPU Average Load	Reciprocity Extraction Time
AutoEncoder [21]	3 h 4 min	89.2%	0.998 ms
KGNet [3]	4 h 58 min	85.7%	1.273 ms
DNN [43]	2 h 36 min	84.5%	0.997 ms
IRS-CRNet	3 h 32 min	90.3%	1.005 ms

## Data Availability

The available DeepMIMO dataset in this study can be found at https://deepmimo.net/scenarios/o1-scenario (accessed on 26 November 2022). In addition, the randomness testing suit used in this paper can be found at [https://github.com/stevenang/randomness_testsuite (accessed on 26 November 2022)].

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
