# Peer review of "Intelligent Reflecting Surface-Assisted Physical Layer Key Generation with Deep Learning in MIMO Systems"

_sensors, 2022, doi:10.3390/s23010055_

Round 1

Reviewer 1 Report

The authors studied intelligent reflecting surface-assisted physical layer key generation with deep learning in MIMO systems, which is a promising technology to establish effective secret keys. However, I have several doubts.

1.       In the abstract and introduction, the authors should give more details about the challenges if we study the PLKG problem in MIMO systems, and How to address these challenges in this paper.

2.       The author needs to conduct a more comprehensive literature survey. There are lots of literature studying the PLKG problem in MIMO systems without IRS, however, in the related works of this paper, the authors did not mention them. Besides, the authors should clearly state the advantages of introducing IRS into MIMO systems.

3.       The definition of the channel model is confusing. According to Eq (1) and Eq (2), the channel includes amplitude response and phase response. However, Eq (4) and Eq (5) only write phase responses without exp. If the results of Eq (4) and Eq (5) are brought into Eq (1) and Eq (2), a huge error will occur. Besides, I think Eq. (5) is not the optimal phase of $\phi$. What’s more, why is the number of antennas fixed at 8?

4.       The authors claim that there is serious interference in the scenario they are studying, but this interference is not reflected in the user achievable rate. In addition, I do not know what variables needed to be solved in this paper and what problems the proposed in-depth learning method should solve.

5.       The vertical axis of many simulation diagrams is named "value", which is inaccurate. The name of the horizontal axis is "index". What does it mean?

6.       The language in the article needs to be improved, and some symbol definitions are confusing, for example, $N$ has multiple meanings.

Reviewer 2 Report

  This paper proposes a IRS-assisted PLKG method which uses deep learning to extract the channel reciprocity in MIMO systems with TDD. As the systems with multiple antennas were not be widely studied, the originality of this paper looks good, and the methods used to improve PLKG performance are well organized to emphasize the advantage of the proposed scheme.

  Thus, the reviewer thinks that this paper is suitable for publication in this current form. One minor suggestion is that it would be much better if the authors can provide more insights for future studies in the conclusion section. The current form of the conclusion section is just a summary of the paper. Moreover, some typos must be edited before publication. 

Reviewer 3 Report

The study proposed a MIMO based IRS assisted communication for physical layer key generation. IRS-CRNet model was proposed, and simulation study was presented. The authors should consider the following comments:

1. Authors need to specify the research gaps and explain with more references how this work is novel.

2. What is the reason for choosing Convolutional network? Authors need to justify this choice. Recurrent neural networks can also be a choice for this application.

3. Complexity of the proposed algorithm should be derived.

4. The authors reported the previous studies can not achieve excellent results, thus authors need to provide a thorough benchmarking with the existing studies.

5. What is the reference of equation 5?

6. How will be the performance of the proposed model without IRS?

7. For the training of the deep learning model what is the input size? In addition, how training and test datasets are divided?

8. The manuscript lacks the details of the training process and hyperparameters are not defined. It is essential to provide this information to reproduce the results. How the optimal hyperparameters were chosen ( i.e. Batch size, optimizer, epochs, etc.)?

8. Table 3 is cited before Table 1 and Table 2. Please rearrange the numbering to cite in a consecutive manner.

9. What is “value” in the figure 9 y-axis?

10. A descript is needed of how Table 4 is generated.

11. Authors are encouraged to proofread the manuscript.

Round 2

Reviewer 3 Report

No further comments.